# Cumulative Risk and Mental Health of Left-behind Children in China: A Moderated Mediation Model

**DOI:** 10.3390/ijerph20021105

**Published:** 2023-01-08

**Authors:** Junmei Xiong, Weiwei Xie, Tong Zhang

**Affiliations:** 1Key Laboratory of Adolescent Cyberpsychology and Behavior (CCNU), Ministry of Education, Wuhan 430079, China; 2Key Laboratory of Human Development and Mental Health of Hubei Province, School of Psychology, Central China Normal University, Wuhan 430079, China; 3School of Psychology, Central China Normal University, Wuhan 430079, China

**Keywords:** dual-factor model of mental health, cumulative risk, coping style, gratitude, left-behind children in China

## Abstract

Based on the dual-factor model of mental health (DFM) and the cumulative risk (CR) model, this study aimed to investigate the impact of CR on left-behind children’s mental health and the underlying mechanism involved, specifically the mediating role of coping style and the moderating role of gratitude in the relationship between CR and mental health. The random cluster sampling method was applied to collect data on CR, coping style, gratitude, life satisfaction, and depression from 705 left-behind children (374 boys, *M*_age =_ 12.20 ± 1.25). The moderated mediation analyses indicated that: (1) the moderated mediation model of CR and depression was significant: coping style mediated the relationship between CR and depression, and gratitude moderated this mediating effect with gratitude strengthening the negative association between CR and coping style; and (2) gratitude moderated the relationship between CR and life satisfaction and it also strengthened the negative association between CR and life satisfaction. The findings suggest that the mechanisms of coping style may differ in the relationships between CR and positive and negative indicators of mental health in left-behind children and that gratitude as a protective factor has limited capacity to buffer the negative effect of accumulated risk. These findings provide evidence for differentiated intervention approaches to promote disadvantaged children’s life satisfaction and depression.

## 1. Introduction

Left-behind children are those with both parents or one parent who have or has moved to prosperous cities to work and therefore, they cannot live together with their parents and have to stay in the household registration area [1]. According to the Ministry of Civil Affairs of the People’s Republic of China in 2020, despite a 7.6% decrease in the number of left-behind children compared to the number of 2018, there were still about 6.44 million left-behind children in China. Many of these disadvantaged children have mental health problems [2]. Previous studies have mainly used the SCL-90 questionnaire, Mental Health Test (MHT), and other relevant mental health tests to assess the mental health of left-behind children [3,4]. Meta-analysis studies indicated that compared with non-left-behind children, left-behind children had more externalizing and internalizing problems [3,4].

However, both the SCL-90 questionnaire and the MHT are based on a “problem perspective” and focus on the negative aspects of the mental health of left-behind children, neglecting the measurement of positive indicators of mental health. Under the influence of positive psychology [5], Greenspoon and Saklofske proposed a dual-factor model of mental health (DFM), which uses subjective well-being (SWB) and psychopathological symptoms to comprehensively assess adolescent mental health [6]. Among the indicators (i.e., life satisfaction, positive affect, negative affect) for SWB, life satisfaction better reflects Chinese people’s true level of well-being than positive affect and negative affect [7]. Furthermore, life satisfaction has a positive effect on adolescent emotional, mental, and social functioning [7], and is a core indicator of adolescent mental health [8]. Thus, based on the theoretical framework of DFM, this study selected life satisfaction as an indicator of left-behind children’s positive mental health. In addition, according to the Report on National Mental Health Development in China (2019–2020), the prevalence of depression among Chinese adolescents is 24.6%, which is one of the most common mental diseases among adolescents. This study, thus, selected depression as an indicator of left-behind children’s negative mental health.

In addition, bioecological theory suggests that individual development is simultaneously influenced by multiple ecological risk factors such as family, school, and community [9]. As risks tend to co-occur, the cumulative risk (CR) approach of measuring the multiple risks, that children and adolescents are exposed to, is used to quantify the number of risks an individual is exposed to [10]. Each risk factor is dichotomously coded (with risk = 1, without risk = 0) according to the screening criteria in the literature, and all risk elements are then aggregated to produce a CR index [10]. As risks tend to co-occur, compared with using single risk to predict youth developmental outcomes [11,12], the CR approach has the advantage of quantifying the accumulated risk adolescents are exposed to, and is therefore helpful to understand the cumulative effect of ecological risk on adolescent mental health [11,12]. There has been ample research on CR and adolescent mental health [13,14,15,16,17]. Nonetheless, there is a lack of research on Chinese left-behind children’s mental health using the CR approach and we are unclear about the role of coping style and gratitude in the relationship between CR and their mental health. Therefore, this study tested the mechanisms involved.

### 1.1. The Mediating Role of Coping Style in the Relationship between CR and Mental Health

Many empirical studies have shown that CR could enhance individuals’ internalizing and externalizing problems and threaten their psychological well-being [13,14,15,16,17]. CR predicted the problematic behaviors of US adolescents one year later (i.e., externalized behaviors and internalized problems) [13]. In another longitudinal study, preschool CR exposure predicted US children’s internalization behaviors when they were 10 years old [16]. Other researchers also found that childhood CR exposure predicted depression in early adulthood among Finnish teenagers [15]. Furthermore, in studies with Chinese adolescents, CR positively predicted psychological distress, anxiety and depression, delinquency, and substance abuse, and negatively predicted life satisfaction [14,17].

Coping style could function as a mediator in the relationship between CR and the mental health of left-behind children. Coping style refers to an individual’s behavioral or psychological efforts to control, reduce, or minimize stressful events [18]. According to the stress and coping theory [19], when individuals face external stimuli, they go through a serial stress process of “internal and external stressors → intermediary variables → physical and mental health”, and coping style is one of the important intermediary variables in this process. Children’s coping styles include positive coping and negative coping [20]. Positive coping is beneficial to solve problems and includes coping strategies such as self-regulation and seeking help from others; negative coping tends to avoid problems and includes coping strategies such as fantasy and escape [21]. When exposed to adverse events, left-behind children may choose different strategies to cope with the threats. The variable of CR examined in this study is the accumulation of a series of stressful life events, which is essentially a stressor of the external environment reflecting multiple threats. As a result, when left-behind children are threatened by CR, their coping style may help them cope with the effect of CR on their mental health. In a longitudinal empirical study, researchers examined the relationship between CR, coping, and adjustment. Specifically, participants were followed for three years and assessed every other year. The results revealed that CR at Time 1 predicted adjustment at Time 2 among preadolescent children through avoidant coping [22]. It is our interest to know whether coping style could also mediate the relationship between CR and adolescent mental health in left-behind children in China.

### 1.2. The Moderating Role of Gratitude in the Relationship between CR and Mental Health

Gratitude refers to a psychological tendency that people use to recognize and respond to other people’s kindness and help in the positive experiences and outcomes that one obtains [23]. Gratitude may moderate the direct pathway between CR and mental health. A meta-analysis of 62 gratitude-related studies found that individuals with higher gratitude reported higher subjective well-being [24]. Gratitude has a protective effect on individual mental health [23,25,26]. There are two kinds (i.e., risk buffering and risk enhancement) of protective roles [11]. Moderators as risk-buffering factors alleviate the negative impact of risk on developmental outcomes, whereas moderators as risk-enhancing factors strengthen the association between risk and developmental outcomes. For instance, gratitude moderated the effects of intrusive rumination on post-traumatic growth in adults, and individuals with high gratitude were able to recover more quickly from the pain [25]. The protective role of gratitude presents itself as a risk-buffering model. In contrast, in a risk-enhancing model, gratitude moderated the effects of incivility on psychological distress among college students, and this moderating effect was only evident at low levels of gratitude [26]. We expect that gratitude would moderate the associations between CR and left-behind children’s mental health (i.e., life satisfaction, depression).

### 1.3. The Moderating Role of Gratitude in the Relationship between CR and Coping Style

Gratitude may also moderate the effect of CR on the coping style among left-behind children. Gratitude is closely related to coping style [27,28]. Wood et al. found that coping style is a psychological resource owned by individuals with gratitude characteristics. Specifically, coping style mediates the effect of gratitude on perceived stress, and gratitude can reduce perceived stress through a positive coping style [27]. In another cross-sectional study, gratitude positively predicted positive coping style and negatively predicted negative coping style; and coping style mediated the effect of gratitude on adolescents’ internalized and externalized problem behaviors [28]. According to the broaden-and-build theory of positive emotions, positive emotional traits can enhance individual cognitive function and broaden individual momentary thought-action repertoires to better cope with adverse events [29]. In this study, gratitude, as a positive emotional trait, may enhance the cognitive functioning of left-behind children to influence their coping style, and individuals with high gratitude may adopt more active coping strategies when faced with exposure to CR.

### 1.4. The Present Study

This study aimed to test the association between CR and mental health (i.e., life satisfaction, depression) among left-behind children in China. Moreover, the mediating role of coping style and the moderating role of gratitude in the relationship between CR and mental health were investigated. The hypotheses of this moderated mediation model (see Figure 1) were tested: (1) coping style will mediate the relationship between CR and mental health (i.e., life satisfaction, depression) among left-behind children in China; (2) gratitude will moderate the relationship between CR and mental health; (3) gratitude will moderate the relationship between CR and coping style.

## 2. Methods

### 2.1. Participants and Procedure

To enhance the findings’ generalizability, the random cluster sampling method was applied to recruit participants from left-behind children in densely populated provinces across China (i.e., one school from Hubei, one school from Henan, and two schools from Inner Mongolia) in March 2018. Informed consent was obtained from the participants and their caretakers or schoolteachers in case participants were unattended by their parents or family members. Permission to conduct the study was obtained from the four schools. Participants were ensured that their data would be confidential. The research team administered a Chinese paper-and-pencil questionnaire to 1600 school students in a classroom setting and 1376 of them filled in questionnaires. The return rate was 86%. It took participants 30 min to complete the survey. A small gift was provided to participants to thank them for their contribution.

After removing invalid questionnaires and participants who were not left-behind children, a total of 705 left-behind children were recruited as the sample of this study. Of the participants, 53% were male, with a mean age of 12.20 years (SD = 1.25). The sample consisted of 5th to 9th graders with 157, 168, 117, 144, and 119 in each grade, respectively. Among the participants, 185 lived in cities, 248 lived in towns, and 272 lived in rural areas. There were 413, 68, and 224 children whose fathers, mothers, and both parents were absent at home, respectively. Furthermore, 381 students (54%) reported that their parents had been away from home for more than 6 months. The left-behind participants were either taken care of by single parents (*n* = 467), grandparents (*n* = 205), or other relatives (*n* = 15) or unattended (*n* = 18).

### 2.2. Measures

#### 2.2.1. CR

In line with the previous research [13,17,28], four domains of risk factors (i.e., family, school, peer, and community) were selected to construct a CR index.

Family risk factors.

Fathers’ and mothers’ educational level was measured using two items [11]. A 7-point scale was used from 1 (never went to school) to 7 (graduate and above). Participants whose two parents did not complete high school were coded as having risk.

Family financial difficulty was assessed using 4 items [30]: “My family does not have enough money to buy clothes”, “My family does not have enough money to buy the food I like”, “My family does not have enough money to buy good housing” and “My family does not have money left over for the family to entertain”. Participants reported the frequency of financial stress in the household in the past year. A 5-point scale (1 = never, 5 = always) was used. Cronbach’s α was 0.849.

Parent–child relationship was measured using the 9-item Closeness to Parents Scale by Buchanan et al. [31], which has sound psychometric properties in Chinese adolescent samples [32]. Children reported their relationship with each parent. The scale ranges from 1 (not at all) to 5 (very). Cronbach’s α was 0.85 (father–child relationship) and 0.84 (mother–child relationship).

To measure the perceived parenting style of the main caregiver of left-behind children, this study used the revised Chinese version of a s-EMBU (short-form Egma Minnen Bardndom av Uppforstran) [33]. This scale has 21 four-point (1 = never, 4 = always) items and has three subscales: rejection, emotional warmth, and overprotection. Cronbach’s α was 0.736 (parenting style of rejection), 0.837 (parenting style of emotional warmth), and 0.635 (parenting style of over protection), respectively.

2.School risk factors.

Teacher–student relationship was measured using the teacher–student relationship questionnaire [34], with a total of 23 items ranging from 1 (not at all) to 5 (very). The scale contains four dimensions: conflict, intimacy, support, and satisfaction. Cronbach’s α was between 0.660 and 0.828.

School connectedness was measured using three items [13], which were used to measure students’ level of belongingness to school. Items range from 1 (completely disagree) to 5 (completely agree). Cronbach’s α was 0.707.

3.Peer risk factors.

Peer relationship was measured using a Chinese version of Friendship Quality Questionnaire created by David [35] and revised by Xi-Xi Cui [36], which consists of 14 items on a 7-point scale ranging from 1 (no at all) to 7 (fully compliant), with a higher score indicating better peer relationship. The Cronbach’s α in this study was 0.910.

4.Community risk factors.

Social support from neighbors was measured using a subscale of Perceived Social Support Scale (PSSS) [37], which contains 4 items, each scored on a 7-point scale (1 = strongly disagree, 7 = strongly agree). Higher scores indicate more social support from others (not including significant others). The Cronbach’s α in this study was 0.88.

According to previous research [12,17,38], the CR score for each participant was calculated by aggregating the risks identified and the 75th or 25th percentile of interval scales was used as the threshold to screen whether participants were exposed to a risk. Financial difficulty, rejection and overprotection of parental rearing behavior, and conflictive teacher–student relationship above or equal to the 75th percentile were coded as 1 (risk) and others as 0 (no risk); parent–child relationship (father/mother), emotional warmth of parental rearing behavior, cohesive teacher–student relationship, social support from neighbors, school connectedness, and peer relationship below or equal to the 25th percentile were coded as 1 (risk) and others as 0 (no risk). The CR score ranged from 0 to 14.

#### 2.2.2. Gratitude

The Gratitude Questionnaire (GQ-6) [23] was used to assess gratitude. We used the Chinese version translated by Zhang Ping [39]. This questionnaire consisted of 6 items, which were scored on 7 points scale ranging from 1 “strongly disagree” to 7 “strongly agree”, with higher scores indicating higher levels of gratitude. The GQ has sound psychometric properties among Chinese adolescents [40,41]. Cronbach’s α was 0.724 in this study.

#### 2.2.3. Coping Style

Coping style was assessed by the 20-item Simplified Coping Style Questionnaire (SCSQ) [20], which was scored on a 6-point scale ranging from 1 (never) to 6 (always). SCSQ was developed to measure Chinese children and adolescents’ coping styles [21]. This scale contains two subscales (i.e., positive coping and negative coping). The overall coping style depends on the magnitude of their positive and negative coping style scores, which is calculated by the mean difference between the standard scores of positive coping style and negative coping style [21]. A total score greater than 0 indicates that individuals tend to adopt a positive coping style; whereas a score less than 0 indicates that individuals tend to adopt a negative coping style. The Cronbach αs were 0.798 (positive coping) and 0.677 (negative coping) in this study.

#### 2.2.4. Mental Health

Life satisfaction was measured using the Chinese version of the revised Students’ Life Satisfaction Scale (SLSS) [42,43], which consists of 7 items and is scored on 6 points ranging from 1 (completely disagree) to 7 (completely agree). Higher scale scores indicate higher levels of life satisfaction. The Cronbach’s α in this study was 0.69.

Depression was assessed using the depression subscale of the Youth Self Report (YSR) of Achenbach′s Behavior Checklist revised by Liu et al. [44]. This scale has a total of 16 items, with a 3-point scale ranging from 0 for “never” to 2 for “always”, with higher scores indicating higher levels of depression. Cronbach’s α was 0.847.

### 2.3. Data Analysis

This study used SPSS Version 26 for statistical analysis of the data. First, because all the data were self-reported, Harman’s single-factor test was used to test the data for common method bias. Then, descriptive statistics of means, standard deviations, skewness, kurtosis, and correlations were performed for the variables in the study. Last, we used model 8 of the PROCESS macro in SPSS 26.0 to perform a moderated mediation effects test [45,46]. Specifically, we constructed two moderated mediation models for life satisfaction and depression. In each moderated mediation model, we first examined the moderating role of gratitude in the relationship between CR and coping style. Then, we tested the moderating role of gratitude and the mediating role of coping style in the relationship between CR and mental health (i.e., life satisfaction and depression). In addition, age, gender, parents’ time away from home, number of parents absent at home, place of residence, and type of care were covariates in this study to exclude their effects on the associations among the variables.

## 3. Results

### 3.1. Preliminary Analyses

The results of the common method biases test revealed that the biggest single factor has an explained rate of 12.83%, which is below the critical value of 40%, indicating that there is no serious common method bias [47]. Next, the means, standard deviations, skewness, kurtosis, and correlation coefficients of all variables were calculated (Table 1). The values of the variables range from −0.28 to 0.98 for skewness and from −0.55 to 0.61 for kurtosis. Since the absolute value of skewness of all variables is less than 2 and the absolute value of kurtosis is less than 7, the variables in this study can be considered as basically conforming to the approximately normal distribution [48]. All study variables were significantly correlated with each other (Table 1).

### 3.2. CR and Depression: A Moderated Mediation Test

Model 8 of Process 4.0 was used to output two equations and the demographic variables were covariates. The first was a regression equation with coping style as the dependent variable, the second was a regression equation with depression as the dependent variable. As was shown in Table 2, CR had a significant negative predictive effect on cope style (β = −0.13, 95%CI = [−0.17, −0.10]) and the negative effect of coping style on the depression of left-behind children was also significant (β = −0.06, 95%CI = [−0.08, −0.03]), suggesting that coping style was a partial mediator between the relationship of CR and left-behind children’s depression. CR had a significant positive predictive effect on left-behind children’s depression (β = 0.04, 95%CI = [0.03, 0.05]) when the moderator variable (gratitude) and the mediator variable (coping style) were included. Furthermore, the interactive item of gratitude and CR on coping style was significant (β = −0.04, 95%CI = [−0.06, −0.01]); whereas the interactive effect of CR and gratitude on depression was not significant (β = −0.003, 95%CI = [−0.01, 0.004]). Thus, gratitude moderated the first half of the indirect path of “CR → coping style → depression” and did not moderate the direct path between CR and depression.

In order to further test how gratitude moderated the association between CR and coping style, a simple slope test was conducted (see Figure 2). For left-behind children with low gratitude (M − 1SD), CR negatively predicted coping style (simple slope = −0.20, *p* < 0.001); for the left-behind children with high gratitude (M + 1SD), CR still negatively predicted coping style, but the predictive effect was stronger (simple slope = −0.41, *p* < 0.001). Results revealed that gratitude strengthened the negative association between CR and left-behind children’s coping style. Gratitude had a relatively weaker protective effect on coping style when left-behind children were exposed to high levels of CR. In other words, gratitude played a risk-enhancing role in the relationship between CR and coping style.

### 3.3. CR and Life Satisfaction: A Moderated Mediation Test

Model 8 of PROCESS 4.0 was used to test the moderated mediation model of CR and left-behind children’s life satisfaction. The results (Table 3) showed that CR negatively predicted coping style (β = −0.13, 95%CI = [−0.17, −0.10]), but coping style did not significantly predict life satisfaction (β = 0.01, 95%CI = [−0.04, 0.07]), indicating that coping style was not a mediator between the relationship of CR and life satisfaction. In addition to the significant interactive effect of CR and gratitude on coping style (β = −0.04, 95%CI = [−0.06, −0.01]), the interactive effect of CR and gratitude on life satisfaction was also significant (β = −0.02, 95%CI = [−0.04, −0.002]), indicating that gratitude not only played a moderating role in the relationship between CR and coping style but also between CR and life satisfaction.

In order to further test how gratitude moderated the association between CR and life satisfaction, a simple slope test was conducted (see Figure 3). Gratitude also strengthened the negative association between CR and left-behind children’s life satisfaction (simple slope = −0.34, *p* < 0.001, gratitude = M − 1SD; simple slope = −0.50, *p* < 0.001, gratitude = M + 1SD). Gratitude had a relatively weaker protective effect on life satisfaction when left-behind children were exposed to high levels of CR. Therefore, gratitude also demonstrated a risk-enhancing effect.

## 4. Discussion

This study offers several unique contributions. First, the previous research on CR and youth mental health [10] has mainly focused on its impact on children and adolescents. This study investigated the association between CR and mental health of left-behind children in China, thus, it expanded the external validity of the CR model in a disadvantaged group. Second, previous research on CR and adolescents’ mental health has primarily focused on psychopathology. This study brings a complete mental health perspective [6] to how life satisfaction and depression relate to cumulative adversity among disadvantaged children.

The present study identified different mechanisms of gratitude and coping style between the associations of CR and different mental health outcomes (i.e., life satisfaction, depression) among Chinese left-behind children. Specifically, the moderated mediation model between CR and left-behind children’s depression was significant with coping style partially mediating the relationship between CR and depression, and gratitude moderating the relationship between CR and coping style. In contrast, the moderated mediation model between CR and left-behind children’s life satisfaction was not significant, as coping style didn’t mediate the relationship between CR and life satisfaction, but gratitude still moderated the relationship between CR and life satisfaction.

### 4.1. The Moderated Mediation Effect of Coping Style and Gratitude in the Relationship between CR and Left-behind Children’s Depression

This study found that coping style partially mediated the relationship between CR and left-behind children’s depression, providing partial support for Hypothesis 1. Consistent with previous research findings [38,49], CR was positively associated with left-behind children’s depression. Researchers found that emotional dysregulation mediated the relationship between cumulative adversity and depression [50]. In this study, left-behind children under high adversity may experience a decrease in positive emotions and an increase in negative emotions and have emotional dysregulation, which could lead to higher level of depression. Furthermore, the findings support the stress and coping theory [19] in left-behind children. According to this theory, coping style is a dynamic process of interaction between the individual and the environment. The mediating role of coping style between stressors (CR) and adaptive outcomes (depression) confirms that coping style is an important tool for left-behind children to cope with adversity and to promote their psychological adjustment. In the present study, CR is essentially a series of stressful life events. Environmental factors and individual differences could lead to different coping styles in response to adverse events, and individual coping styles could further influence their psychological well-being. A meta-analysis of a sample of 147,523 Chinese showed that mental symptoms were negatively associated with positive coping style and positively associated with negative coping style [51]. This suggests that fostering positive coping should be considered when designing prevention and intervention programs for decreasing left-behind children’s depression.

The first half of the mediated pathway “CR → coping style → depression” was moderated by gratitude. The significance of the moderated mediational model of CR and left-behind children’s depression provided support for Hypothesis 3. The coexistence of the mediating role of coping style between the association of CR and depression and the ‘risk-enhancing’ moderating role of gratitude in the relationship of CR and coping style suggested that the development of individual resilience under severe adversity comes at a cost. Although coping style could mediate the negative effect of CR on left-behind children’s depression, gratitude was effective in protecting coping style at low CR and this effect was weakened at high CR. The weakened protective role of gratitude in coping style under high adversity might be the “cost of resilience” for left-behind children when they try to cope with environmental adversity [11,52]. This finding might be specific to the children in the Chinese culture. In the Western individualistic culture, gratitude is more closely related to positive feelings [53], whereas in Eastern collectivism culture, Confucianism plays an important role in interpersonal relationships; for example, gratitude itself brings a sense of indebtedness [54]. Left-behind children with high gratitude are grateful for the caring and support from their parents, teachers, and friends and therefore feel strong obligations to give them something in return. If they are incapable of rewarding the giver, individuals with high gratitude would develop a great sense of failure and frustration [55]. The sense of indebtedness can be very stressful for the recipient, making it counterproductive [56]. Thus, the psychological burden of not being able to repay the benefactor at high risk may offset the positive effects of gratitude and could eventually lead to more negative coping.

### 4.2. The Moderating Effect of Gratitude between CR and Left-behind Children’s Life Satisfaction

We also identified a ‘risk-enhancing’ moderating role of gratitude in the relationship between CR and left-behind children’s life satisfaction. On the one hand, left-behind children with high gratitude demonstrated higher levels of life satisfaction when facing the threat of CR compared to those with low gratitude. This may be due to the predictive effect of gratitude as an important asset for one’s life satisfaction [57]. Children with high gratitude focus more on the positive aspects of life and look forward to returning the help and kindness of others. On the other hand, the moderating role of gratitude in the relationship between CR and life satisfaction was “risk-enhancing”. Although gratitude could buffer against the negative effect of CR on left-behind children’s life satisfaction, the protective role of gratitude was weakened at high levels of CR. Accordingly, when school psychologists design prevention and intervention programs to facilitate children and adolescents’ life satisfaction through fostering their gratitude, they should keep in mind that the protective role of gratitude in life satisfaction is limited especially when children are exposed to high adversity and a more comprehensive program nurturing positive attributes such as gratitude and decreasing the amount of ecological risk would be more likely to promote healthy development among left-behind children.

Furthermore, contrary to previous research [58], the present study did not identify the mediating role of coping style in the relationship between CR and life satisfaction. This inconsistency may be due to the mediating role of coping style and the moderating role of gratitude being considered when analyzing the association of CR and life satisfaction. The mediating effect of coping style may be exactly the opposite at different levels of gratitude, resulting in an insignificant mean when aggregating the mediating effects. Accordingly, gratitude was removed from the model, but the mediating effect of coping style was still not significant (β = −0.007, 95% CI [−0.017, 0.029]), suggesting that CR does not affect left-behind children’s life satisfaction through coping style. Some scholars have found that withdrawing coping strategies instead of positive coping could mediate the effect of stress on adolescents’ life satisfaction [59]. This suggests that under severe adversity, although coping style may temporarily help children deal with challenges and decrease their depression, it does not reduce their dissatisfaction with their current life and increase their hope for a better life in the future.

### 4.3. Limitations and Suggestions for Future Research

This study has several limitations. First, all the data were collected via a self-report method and social desirability might be a concern. Parents’ and teachers’ reports on CR and adolescent mental health should also be collected to increase the validity of the study. Second, this study implemented a cross-sectional design, which does not allow us to make causal statements between and among the variables. Follow-up studies can adopt a longitudinal design to collect data at multiple time points so that the predictive effect of CR on adolescent mental health and the mechanism involved could be verified. Third, GQ-6 was originally developed to measure adults’ gratitude and later revised to measure Chinese adolescents’ gratitude [40]. The revised GQ-6 for adolescents has been tested to be a reliable tool to measure Chinese adolescents’ gratitude. Nonetheless, children and adolescents might not experience gratitude in the same way as adults [53]. Furthermore, the items 5 (i.e., As I grow older, I am understanding more and more the help from the persons, experiences, and things in my life, which are all part of my life history) and 6 (i.e., It takes a long time before I feel grateful for someone or something) of the GQ-6 might be more applicable to adult experiences. Therefore, more relevant studies are needed to test the validity of the GQ-6 in adolescent samples in different cultural contexts. Finally, we did not include a comparative group in our study and therefore could not answer whether the associations among the variables were identical or not in non-left-behind children. Future studies could include a comparative group besides the target group so that the mechanisms of cumulative risk effects on children’s mental health could be explored further.

## 5. Conclusions

Gratitude and coping style have different functioning in the relationship between CR and left-behind children’s mental health (i.e., depression, life satisfaction). In the association of CR and depression, coping style has a mediating role in this relationship and gratitude moderates the relationship between CR and coping style, and the protective effect of gratitude is weakened when the environmental adversity is high. In the association of CR and life satisfaction, coping style does not have a mediating role in this relationship and gratitude moderates the relationship between CR and life satisfaction, and the protective effect of gratitude is weakened when environmental adversity is high. The different mechanisms of gratitude and coping style in the association between CR and left-behind children’s mental health (i.e., depression, life satisfaction) provide empirical support for designing and implementing differentiated programs to decrease depression and increase life satisfaction for disadvantaged children. When disadvantaged children are exposed to relatively low environmental adversity, cultivating their gratitude, and facilitating their coping style is helpful to decrease their depression; whereas facilitating their gratitude is helpful to increase their life satisfaction. Nonetheless, we shall not overestimate the protective effect of gratitude when disadvantaged children are exposed to high adversity as its protective effect is limited.

## Figures and Tables

**Figure 1 ijerph-20-01105-f001:**
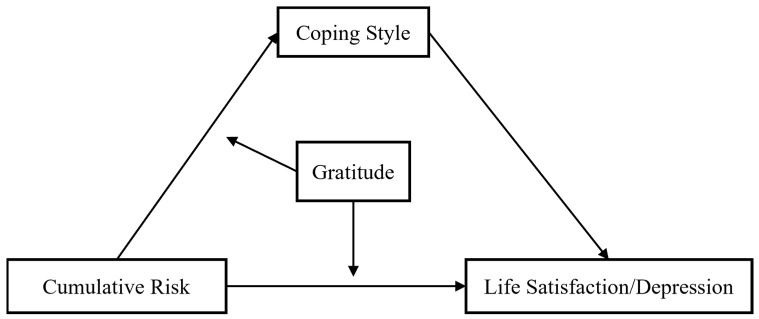
A moderated mediation model.

**Figure 2 ijerph-20-01105-f002:**
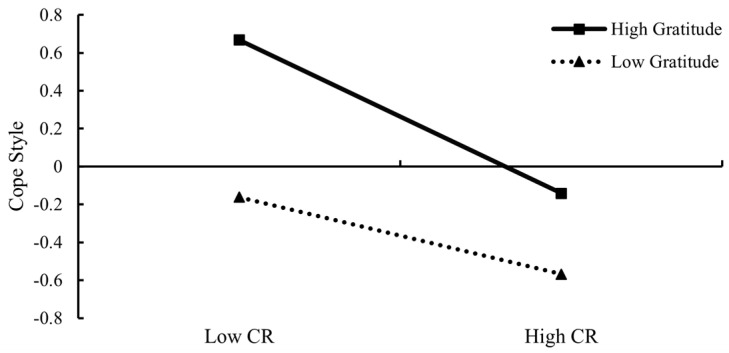
The interaction of CR and gratitude on cope style. Note: CR = cumulative risk.

**Figure 3 ijerph-20-01105-f003:**
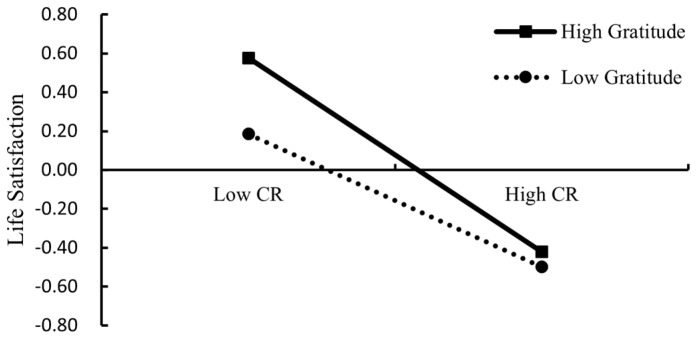
The interaction of CR and gratitude on life satisfaction. Note: CR = cumulative risk.

**Table 1 ijerph-20-01105-t001:** Descriptive statistics and correlations for the main variables (*n* = 705).

Variable	M	SD	Skew	Kurt	1	2	3	4
1. CR	4.20	2.85	0.64	−0.24				
2. Gratitude	5.17	1.08	−0.23	−0.55	−0.50 ***			
3. Cope tendency	0.00	1.23	0.06	−0.05	−0.44 ***	0.45 ***		
4. Life satisfaction	3.62	0.86	−0.28	0.28	−0.46 ***	0.32 ***	0.25 ***	
5. Depression	1.42	0.34	0.98	0.61	0.37 ***	−0.21 ***	−0.31 ***	−0.28 ***

Note: *** *p* < 0.001.

**Table 2 ijerph-20-01105-t002:** Test for moderating mediating effects of CR on depression.

Predictive Variables	Outcome Variable: Cope Style			Outcome Variable: Depression		
	β	t	95%CI	β	t	95%CI
Constant	−0.71	−1.17	[−1.91, 0.48]	0.48 **	2.71	[0.13, 0.83]
Age	−0.05	1.15	[−0.03, 0.12]	−0.01	−0.69	[−0.03, 0.01]
Sex	0.12	1.51	[−0.04, 0.28]	0.06	2.61	[0.02, 0.11]
Place of residence	−0.04	−0.72	[−0.16, 0.08]	−0.05 *	−2.57	[−0.08, −0.01]
Number of parents Absent at home	−0.06	−1.06	[−0.18, 0.05]	0.01	0.83	[−0.02, 0.05]
Time away from home	0.03	0.41	[−0.13, 0.19]	0.01	0.57	[−0.03, 0.06]
Type of care	0.05	0.59	[−0.12, 0.22]	−0.01	−0.33	[−0.06, 0.04]
CR	−0.13 ***	−8.05	[−0.17, −0.10]	0.04 ***	7.25	[0.03, 0.05]
Gratitude	0.34 ***	7.80	[0.25, 0.43]	0.01	0.41	[−0.02, 0.03]
CR* Gratitude	−0.04 **	−3.12	[−0.06, −0.01]	−0.00	−0.81	[−0.01, 0.00]
Cope style				−0.06 ***	−4.96	[−0.08, −0.03]
R^2^	0.28			0.18		
F	30.38 ***			15.70 ***		

Note: N = 705. Bootstrap sample size = 5000. CI = confidence interval. * *p* < 0.05; ** *p* < 0.01; *** *p* < 0.001. CR = cumulative risk.

**Table 3 ijerph-20-01105-t003:** Test for moderating mediating effects of CR on life satisfaction.

Predictive Variables	Outcome Variable: Cope Style			Outcome Variable: Life Satisfaction		
	β	t	95%CI	β	t	95%CI
Constant	−0.71	−1.17	[−1.91, 0.48]	3.48 ***	7.97	[2.62, 4.34]
Age	−0.05	1.15	[−0.03, 0.12]	0.04	1.35	[−0.02, 0.09]
Sex	0.12	1.51	[−0.04, 0.28]	−0.05	−0.79	[−0.16, 0.07]
Place of residence	−0.04	−0.72	[−0.16, 0.08]	0.02	0.47	[−0.07, 0.11]
Number of parents absent at home	−0.06	−1.06	[−0.18, 0.05]	0.00	0.06	[−0.08, 0.09]
Time away from home	0.03	0.41	[−0.13, 0.19]	−0.14 *	−2.41	[−0.25, −0.03]
Type of care	0.05	0.59	[−0.12, 0.22]	−0.09	−1.45	[−0.21, 0.03]
CR	−0.13 ***	−8.05	[−0.17, −0.10]	−0.13 ***	−10.11	[−0.15, −0.10]
Gratitude	0.34 ***	7.80	[0.25, 0.43]	0.09 **	2.64	[0.02, 0.15]
CR * Gratitude	−0.04 **	−3.12	[−0.06, −0.01]	−0.02 *	−2.22	[−0.04, −0.00]
Cope style				0.01	0.47	[−0.04, 0.07]
R^2^	0.28			0.24		
F	30.37 ***			22.53 ***		

Note: *n* = 705. Bootstrap sample size = 5000. CI = confidence interval. * *p* < 0.05; ** *p* < 0.01; *** *p* < 0.001. CR = cumulative risk.

## Data Availability

The datasets generated during and/or analyzed during the current study are available in the Ann Arbor, MI: Inter-university Consortium for Political and Social Research (ICPSR) repository (https://doi.org/10.3886/E183302V1 (accessed on 29 November 2022).

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
