# Peer review of "Cumulative Risk and Mental Health of Left-behind Children in China: A Moderated Mediation Model"

_ijerph, 2023, doi:10.3390/ijerph20021105_

Round 1

Reviewer 1 Report

 The research paper is based upon an interesting study which investigated nature of relationship between some significant psychological variables in vulnerable population.  Some of the comments are made to improve the clarity and consistency while reporting methodology. There is need to achieve more accuracy in use of appropriate terms and expression when discussing findings. 

Introduction:

The introduction section is describing the study variables and their interrelation adequately. However, some minor adjustments are needed for instance (Line 94-99) mentions about technique used by authors to measure the coping style that relates more to the method section rather introduction, and it also seems somewhat disjointed with previous and following paragraphs. The authors didn’t mention anything about positive coping or negative coping before and sudden appearance of these ideas that “the overall coping style of individuals depends on the magnitude of their positive and negative coping style scores, which is calculated by the mean difference between the standard scores of positive coping style and negative coping style”. I would prefer revising it by referring to what is positive and negative coping especially with regard to children population here.  The information related to scoring can be moved to method section.

Methods:

The administration of questionnaire procedures not clear.  “A total of 1600 questionnaires were administered and 1376 questionnaires were returned, with a return rate of 86%.” Were the questionnaires self-administered or they were administered by someone because in later section it is mentioned “a paper-and-pencil questionnaire was administered in a classroom setting. The primary examiner was a graduate student in psychology”.

Please be consistent in reporting the data collection procedure and outcome.

The target population for the study are children and adolescents from 5th to 9th grade, and there were (n=18) were unattended. Were all of them capable of providing informed consent? In case if the respondent did not have this capacity, what approach was adopted.

Were all the scales administered in English language, considering the study was completed in schools located in China?

Moreover, were the tools such as The Gratitude Questionnaire (GQ-6) and Coping Styles Questionnaire adapted in accordance with target population?

It is mentioned that “Life satisfaction was measured using the Chinese version of the revised Students’ Life Satisfaction Scale (SLSS) but no such information is given for above-mentioned scales.

Results:

Satisfactory

Discussion

This expression doesn’t seem appropriate “typically developing children and adolescents”. Might need some better synonym for ‘developing’.

While discussing the findings you need to refer to whether children and adolescents experience gratitude in the same way as adults, because some of the items on the scale more relate with adult experiences of gratitude such as item 5 and 6.  You also need to refer to some broader and specific cultural factors in Chinese society that might have role in gratitude and coping mechanisms adopted by children.

One of the limitation of study might be lack of comparative group. A matched control group could provide more insight about the nature of relationship between study variables by comparing its pattern in both groups.

Conclusion:

One of the study implication as reported by authors is “to decrease the number of accumulated ecological risk that disadvantaged children are exposed to”.  However, keeping in view the socioeconomic downfall of the communities at global, regional and national level, some of the risks are inherent. What could be some of the implications related to coping? Coping is an important life skill and why it wasn’t found beneficial in this scenario.   

Author Response

Dear Reviewer,

We are grateful for your constructive comments and suggestions. We have responded to your comments in the attachment.

Sincerely,

Authors

Reviewer 2 Report

The authors dealt an important and interesting topic on the cumulative risk and mental health of left-behind children in China. The mediating role of coping style and the moderating role of gratitude were found in the relationship between cumulative risk and mental health by using the moderated mediation model. The methods utilized are well established, and a cross-sectional survey with over 700 valid questionnaires were conducted. The findings in this study provide some evidences reminding us the importance of promoting disadvantaged childrens life satisfaction and depression. However, there are still some major concerns about this article, and I hope the authors can respond.

I suggest the authors to conduct the sensitive analyses on the influencing factors and cumulative risks. Moreover, according to literature, there are some other environmental factors impacting the childrens mental health. How to adjust these factors in the model adopted in this study?

 It is better to explain more about the representativeness and feasibility to support the study on mental health, like life satisfaction. Which is one of innovation and important finding in this study.

Author Response

(The authors gave the same response as above.)
